# An Approach to Networking a New Type of Artificial Orthogonal Glands within Orthogonal Endocrine Neural Networks

**Miroslav Milovanović [1], Alexandru Oarcea [2,\*], Saša Nikolić [1], Andjela Djordjević [1] and Miodrag Spasić [1]**

1   Faculty of Electronic Engineering, Department of Control Systems, University of Niš, 18000 Niš, Serbia; miroslav.b.milovanovic@elfak.ni.ac.rs (M.M.); sasa.s.nikolic@elfak.ni.ac.rs (S.N.); andjela.djordjevic@elfak.ni.ac.rs (A.D.); miodrag.spasic@elfak.ni.ac.rs (M.S.)
2   Faculty of Automotive, Mechatronics and Mechanics, Department of Mechatronics and Machine Dynamics, Technical University of Cluj-Napoca, 400641 Cluj-Napoca, Romania
\*   Correspondence: alexandru.oarcea@mdm.utcluj.ro; Tel.: +40-722420095

**Abstract:** Currently, artificial intelligence and intelligent algorithms for the control of dynamic systems are the main focus for building Industry 4.0 services and developing novel, innovative industrial solutions. This paper proposes a novel intelligent control structure specifically tailored for treating environmental stimuli and disturbances in operational environments of dynamic systems. The structure is based on the Orthogonal Endocrine Neural Network (OENN) and Artificial Orthogonal Glands (AOGs). The operational mechanism of each AOG acquires and processes environmental stimuli and generates artificial hormone concentration values at the gland output. These values are introduced to the appropriate OENN layer to provoke the network with collected environmental insights. To verify the applicability of the proposed structure on a complex dynamical nonlinear system, it was tested in a laboratory environment on the laboratory magnetic levitation system (MLS). The main experimental goal was to test the tracking performance of a levitation object when the new control logic was applied. The results were compared with two additional intelligent algorithms and a default linear quadratic (LQ) control logic. OENN + AOG structure showed improved tracking performances compared with traditional LQ control and better adaptability to environmental conditions compared with similar existing solutions.

**Keywords:** artificial gland; orthogonal function; endocrine neural network; control logic; magnetic levitation; nonlinear dynamical system

## 1. Introduction and Initial Considerations

### 1.1. State-of-the Art

Real-world systems are often very complex in nature—inherently nonlinear, highly dimensional, time-varying, and, thus, difficult to model or control within the bounds of classical control theory. The solution can often be found in advanced control, i.e., intelligent control techniques [1], which have flourished in the last few decades, especially after breakthroughs in formal mathematical tools related to the optimality, stability, and uncertainty of these systems. These techniques found their applications in various areas such as industry [2], energy [3], or robotic systems [4]. We consider the control method "intelligent" if it employs some artificial intelligence computing approach such as neural networks, fuzzy logic, machine learning, evolutionary computation, or genetic algorithms. Intelligent control may often imply combining two or more artificial intelligence tools into hybrid intelligent systems, such as neuro–fuzzy control, which merges human-like reasoning of fuzzy logic with the learning capabilities and connectionist architecture of neural networks. The most valuable approach to intelligent control proved to be the one based on the application of neural networks, due to a combination of their features such

as parallel processing, adaptability and power of learning. It has been proven that neural networks with nonlinear differentiable activation functions can approximate arbitrary functions, so that recurrent networks were applied in system identification when a set of input-output data pairs was given. Consequently, neural networks were successfully employed in many control algorithms where an accurate model of the controlled system was needed. Paper [5] provided a systematic classification of neural-network-based control. The main thesis was that all approaches could be classified into two categories: a neural network which was auxiliary to the main control logic, and a neural network which was the main controller. In [6], different aspects of stable adaptive neural network control were discussed. Concrete interesting examples of neural network-based control can be found in [7], where a neural network was applied in the control of the four-rotor helicopter, and [8], in the case of the autonomous underwater vehicle.

To overcome major drawbacks in using neural networks for modelling and control of dynamical systems (such as setting the optimal neural network structure, initial values, and training method), the authors of this paper experimented intensively with structures called orthogonal polynomial neural networks [9]: single-layer neural networks based on orthogonal functions (Gegenbauer orthogonal functions [10] are used in this paper). These structures proved superior to classical neural networks in terms of accuracy/complexity ratio due to their natural in-built optimality [11]. Designed orthogonal polynomial neural networks were successfully applied in modeling [12] and control [13] of dynamical systems.

### 1.2. Previous Work of the Authors

Further improvement in the architecture of polynomial neural networks can be achieved by introducing an adaptive measure of variability, which can be used to model uncertainties or time-varying behavior of a real-time operating dynamical system. As this algorithm reminded us of a biological process of glands in the human body producing hormones, it was called the endocrine mechanism. The endocrine mechanism allows the network to adapt to the change in environmental factors, sensed disturbances, measurement errors, and occurred uncertainties within dynamic systems. The authors of this paper proposed numerous solutions based on the utilization of orthogonal functions within endocrine neural networks. Table 1 lists published research, comments regarding their utility value, and contributions to the field of control systems and artificial intelligence.

**Table 1.** A review of the authors' research concerning OENNs.

| Ref. No. | Control Logic | Comment | Contribution |
|---|---|---|---|
| [14] | Orthogonal endocrine adaptive neuro–fuzzy inference systems | The basis of ANFIS is upgraded using orthogonal functions, and the mechanism based on artificial hormonal actions is exploited. | Can be effectively used for modeling dynamic systems. |
| [15] | An orthogonal endocrine neural network-based estimator | An estimator of the modeling error based on the orthogonal endocrine neural network. The orthogonal ANFIS structure enables the dynamical system to adapt to environmental changes. | Optimal steady-state performances for nonlinear discrete-time systems are provided. |
| [16] | A hybrid intelligent structure based on two adaptive neural networks (OENN and OEANFIS) | The main substructure is OEANFIS. Both substructures include Chebyshev orthogonal functions and are empowered with environmental stimuli. | Online tuning of a PID controller and its parameters. |

**Table 1.** *Cont.*

| Ref. No. | Control Logic | Comment | Contribution |
|---|---|---|---|
| [17] | Intelligent real-time controller based on OENN and OEANFIS | The structure uses offline and online learning approaches simultaneously. Additionally, two different types of glands are used: excitatory and inhibitory. | Stable control of nonlinear multiple-input–multiple-output dynamic systems influenced by high-intensive environmental disturbances. |
| [18] | Nonlinear autoregressive neural network (NARX) with activation functions based on orthogonal polynomials | Standard activation functions within the network were substituted with newly developed orthogonal polynomial functions. | A method for improvement of magnetic levitation performances in terms of achieving specified levitation amplitude. |
| [19] | Orthogonal endocrine neural network based on postsynaptic potentials | Artificial inhibitory and excitatory processes are proposed to mimic the self-adjusting biological processes of keeping desired parameters within optimal and specified values. | The architecture is suitable for data forecasting purposes. |

*1.3. Motivation and Argumentation*

The motivation for proposing a novel structure in this paper is threefold. First of all, the idea is to offer an intelligent solution capable of treating numerous environmental stimuli, noises and disturbances, and use the acquired insights to improve the control performances of dynamical systems. The second goal is to make further progress in the domain of artificial glands and their integration within neural networks. Specifically, the aim is to propose a new type of artificial gland characterized by a novel mechanism for treating input signals and generating optimal output values. The final objective is to consider and discuss potential principles of networking of proposed artificial glands with the point of providing extensive stimulation capabilities of underlying neural networks and making a proper framework for merging all environmental stimuli in a single network, enabling each of them to create influence on another stimulus (as in the nature). The concluding task is to test the aforementioned research goals in a real-world environment, by controlling a complex nonlinear dynamical system in a laboratory environment. One of the most popular benchmark systems for trying new control algorithms is the magnetic levitation system (MLS). This system aims to keep a ferromagnetic ball levitating in a place (or to follow a desired trajectory) by applying an adequate voltage to an electromagnet. MLS is a highly nonlinear system with open-loop instability and time-varying dynamics, making it ideal for testing complex control algorithms, and will also be used as a case study for testing the performance of our control algorithm in this paper. The system was not time-varying at the time of operation and individual performance recording during one experimental session. Nevertheless, the system shows a time-varying nature in our long-term work with it. In any experimental attempt, when stable levitation of a ferromagnetic ball is not established and the equilibrium state is not reached, the levitation object moves uncontrollably. It hits the electromagnet, causing the ball to fall out of the system environment and suffer potential damage. Since it is a laboratory model, it does not include protective equipment that would prevent damage to electromagnets and levitation balls. As a result, there are long-term changes in performance due to such damages to the mentioned mechanical elements. Comparing the performance of this specific ML system from a few years ago with the moment of writing this paper, the difference of when the identical experimental setup was made is significant. Described time-varying properties of the system strictly enforce this performance difference. Several exciting approaches emerged over time based on applying different neural networks in the control of MLS. In [20], hybrid control using a recurrent neural network (RNN) was proposed. The controller itself was based on feedback linearization, while RNN functions were used for uncertainty observers. Study [21] considered a neural adaptive method for position control

and identification of MLS. The controller consisted of three parts: PID controller, radial basis function network controller and radial basis function network identifier. On the other hand, a synergistic combination of neural network and sliding mode control was proposed in [22].

Justification for introducing and using this complex, intelligent structure can be found in the following considerations. Generally speaking, artificial neural networks possess the capability of a precise generalization and show robustness when dealing with disturbances. However, they can have difficulties coping with uncertainties, providing interpretability and identifying the importance of each input signal and environmental stimuli for a related system. The last-mentioned deficiency is one of the crucial reasons for considering the OENN + AOG structure for implementation. If an examined dynamical system is linear or simple nonlinear, it cannot be advised to apply a novel structure. Additionally, if there is a lack of data for building a data set, available information is limited or environmental stimuli are not measurable, it is also better to utilize another simpler intelligent or traditional controller. It is justified to use the proposed OENN + AOG structure when it is required to control a complex nonlinear dynamical system when traditional control logics do not provide satisfying results. Particular interest in using the architecture is when numerous measurable environmental stimuli and disturbances exist and can be sensed. The structure's benefit is in introducing an unlimited number of such stimuli to the gland network and later to a designed OENN. It was experimentally shown in the previous section that the novel structure can provide representative results (on the level of the similar complex smart solution) but with significantly less computation time. All acquired and processed environmental stimuli provide a significant value to a connected neural network and controlled system. They can help in training free parameters if treated correctly according to the current environmental conditions.

### *1.4. The Structure of the Paper*

The remainder of the paper is structured as follows. Section 2 presents a novel mechanism of work of artificial glands and the way of processing two different types of input signals. Section 3 introduces an algorithm for networking artificial glands and explains the approach of integrating such a network within one kind of artificial neural network. A universal method of combining the novel intelligent structure within a dynamical system control logic is provided in Section 4. Section 5 introduces a case study based on exploiting a laboratory magnetic levitation system, as the system that will be used for real-time verification of control performances of the proposed system. Finally, in Sections 6 and 7 the experimental results, comparisons of control performances with other already verified control logics and related discussions are presented. In the end, all collected experimental results are used to make generalized conclusions and recommendations toward the applicability of novel intelligent structures.

## 2. Artificial Orthogonal Gland (AOG) Mechanism

This section will present a novel approach to developing artificial glands where a novel artificial orthogonal gland (AOG) mechanism will be proposed. Additionally, a strategy for building a network of AOGs within a single system will be provided. In previous attempts to develop artificial glands [14–16,23–26], the main idea was to acquire environmental stimuli, transform them into appropriate input signals and introduce them to the glands. The next part calculated hormone concentration within each gland and introduced calculated values to a neural network. Each gland was mostly independent of other involved glands, covering different environmental stimuli.

The novel approach proposes a gland structure based on some previously established gland features in combination with new in-gland mechanisms. Further networking of single gland units will enable mutual interactions between different hormones and an additional level of adaptability of connected neural networks. In many previous attempts [14–16,23–26], each environmental stimulus was assumed to be completely inde-

pendent of other stimuli. However, in practice and real-world industrial applications, one environmental condition could affect other related dependencies and volatility of numerous factors. For example, a strong wind signal can significantly affect a control system's performance, if mounted outside. Additionally, the wind can directly influence decreasing the system's temperature, increasing mechanical parts' vibrations and noise level, etc. From this example, it is evident that one environmental stimulus may affect the values of other stimuli. For that reason, a network of hormones can better represent similar situations, simulate mutual correlations, and put the importance and influence of all included stimuli on a system of interest in proper order. In order to propose upgraded functionalities to an artificial gland, the novel gland architecture is proposed in Figure 1.

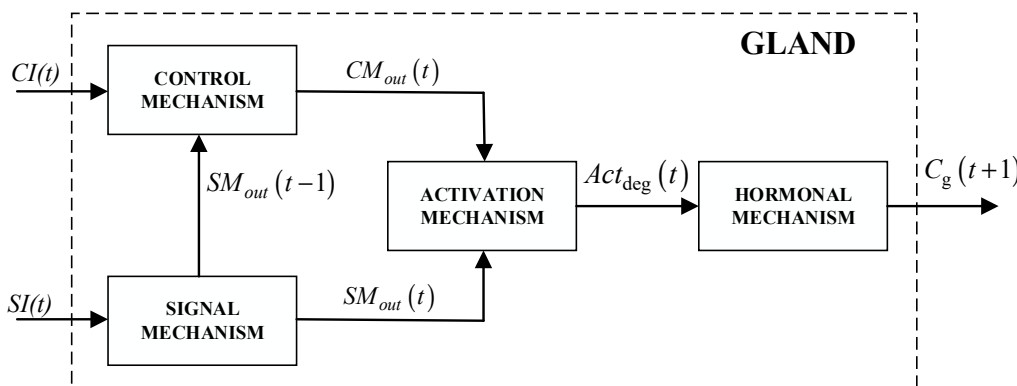

**Figure 1.** AOG mechanism.

The gland mechanism from Figure 1 is designed in such a manner to accept two inputs: control input (*CI*) and signal input (*SI*). The primary purpose of *CI* is to control hormone production, and it is not related to any calculation of hormone stimulations. *CI* enables optional interactions between different glands and allows hormones to connect and make a network of hormones. The second input to the gland mechanism, *SI*, determines the hormone stimulation level within a gland and defines the gland's influence on the neural network. The AOG mechanism includes four main sub mechanisms: control mechanism, signal mechanism, activation mechanism and hormonal mechanism. All these sub-mechanisms will be presented next.

*2.1. Signal Mechanism*

The signal mechanism within the gland cell is used to process environmental signals in the form of SIs and stimulate hormone production. Such a mechanism theoretically enables multiple inputs (environmental stimuli) to access the single gland in comparison with the previous proposals of the authors [17–19]. The output of the signal mechanism can be calculated as:

$$SM_{out}(t) = round(f_{SM}(SI_1(t), SI_2(t), \ldots, SI_n(t))) \tag{1}$$

In (1), *n* is the number of signal inputs introduced to the gland, and $f_{SM}$ function is the standard deviation function. This function shows how dispersed the data of included environmental stimuli are concerning the mean. A low standard deviation implies that the data are clustered around the mean, and a high standard deviation shows that the data are more spread out. If a high deviation occurs, the $SM_{out}$ will be prominent and hormone stimulation will be significant. If dispersion between SI signals is low, $SM_{out}$ will be a small integer value, and its influence on the upcoming gland mechanism will be smaller. The rounding part of Equation (1) is performed to prepare $SM_{out}$ to be applicable within the activation mechanism, which requires integer numbers as inputs.

### 2.2. Control Mechanism

The processing unit of *CI* is the control mechanism (Figure 1). *CI* input can have four different states:

(1)  *CI* = 0 or *CI* = 1: zero-state (inhibitory state) is activated when the gland threshold is not achieved ($0 \leq CI <$ threshold_value), implying a gland not to be active;
(2)  *CI* = 2: when the excitatory manner of work is activated;
(3)  *CI* = 3: when the negative feedback logic of work is activated;
(4)  *CI* = 4: when the positive feedback logic of work is activated.

The state of inhibition (a zero state) is active when the gland threshold is not achieved and results in the output of the control mechanism equal to $CM_{out}(t) = 0$. The threshold value can be set in accordance with control needs: if there is a requirement to build an extra sensitive system that will register even the most minor environmental disturbances, the threshold_value should be set to a value close to 0. If the requirement is to provide a robust system that will activate glands on some significant disturbances or peak occurrences, the threshold_value should be large.

The following introduced feature is the excitatory feature, active when threshold_value is achieved and can be described with the following equation:

$$CM_{out} = f_{CM}(SM_{out}(t-1)). \tag{2}$$

$fCM$ from Equation (2) can be selected as a linear function, bipolar, sigmoid, step, etc. The selection should be made according to a specific use case. The third possibility that the control mechanism can exploit is a negative feedback logic that can be described as follows:

$$CM_{out}(t) = f_{CM}\left(\frac{SM_{out}(t-1)}{1 + CI(t)}\right). \tag{3}$$

When the negative feedback feature (Equation (3)) is active, the hormone production will be suppressed. The final option is to use the positive feedback logic to calculate $CM_{out}$:

$$CM_{out} = f_{CM}\left(\frac{SM_{out}(t-1)}{1 - CI(t)}\right). \tag{4}$$

When the positive feedback feature (Equation (4)) is active, the hormone production intensifies. Negative (Equation (3)) and positive (Equation (4)) feedback controls should be activated once the specified hormone concentrations are achieved (low or high), and the equilibrium points are reached. Feedback controls can rapidly increase or decrease hormone stimulations, enabling quick responses of artificial glands to changes occurring in the environment. Traditional artificial glands need a significant number of iterations to adjust their parameters to sudden changes in environmental conditions and large volatilities. Negative and positive feedback features can make these adjustments significantly faster.

### 2.3. Activation Mechanism

The task of the activation mechanism from Figure 1 is to determine the production of hormone based on $CM_{out}$ and $SM_{out}$. The mechanism processes two signals in each time step, resulting in $Act_{\deg}(t)$:

$$Act_{\deg}(t) = f_{AM}(CM_{out}, SM_{out}), \tag{5}$$

where $f_{AM}$ could be implemented in the form of conventional neural network functions such as sigmoidal, linear, step, Gaussian, etc. However, keeping in mind the previous work of the authors of this paper and extensive experience in developing and implementing novel types of orthogonal functions within neural network environments, this research makes an additional effort in implementing $f_{AM}$ in the form of Gegenbauer orthogonal functions. Gegenbauer functions are orthogonal polynomials defined on the interval $[-1, 1]$. They

generalize Legendre polynomials and Chebyshev polynomials and represent exceptional Jacobi polynomials cases. Gegenbauer polynomials are characterized by rapid convergence rates providing an excellent foundation for building an efficient activation mechanism. Each artificial gland has its row of the orthogonal function: for n glands assigned to the neural network, *n* Gegenbaurer functions will be expended from the initial one and be used for each function (*j* = 0, 1, 2, ... , *n* − 1). Gegenbauer polynomials and appropriate mathematical foundation will be introduced in the following subsection.

### 2.3.1. Gegenbauer (Ultra Sphere) Polynomials

The Gegenbauer polynomials represent a generalization of the classic Legendre polynomials that will be denoted with $C_r^\lambda$, where r is an integer and $\lambda$ is a real number. These polynomials represent the exceptional cases of Jacobi polynomials, and the following relation defines them:

$$\left(1 - 2tx + x^2\right)^{-\lambda} = \sum_{r=0}^{\infty} C_r^\lambda(t) x^r \tag{6}$$

By introducing binomial expansion of the left side of Equation (6), the previous equation can be presented as:

$$(1 - (2t - x)x)^{-\lambda} = \sum_{n=0}^{\infty} \frac{(\lambda)_n x^n (2t - x)^n}{n!} = \sum_{n=0}^{\infty} \sum_{m=0}^{n} \frac{(-1)^m (\lambda)_n (2t)^{n-m} x^{n+m}}{m!(n-m)!} \tag{7}$$

If $k = m \geq 0$ and $r - k = n$, then $n - m = r - 2k$ and $k$ can vary in the range from 0 to [*r*], where:

$$[r] = \begin{cases} \frac{r}{2}, & \text{if } r \text{ is even} \\ \frac{r-1}{2}, & \text{if } r \text{ is odd} \end{cases} \tag{8}$$

Hence, it can be further concluded:

$$(1 - (2t - x)x)^{-\lambda} = \sum_{r=0}^{\infty} \sum_{k=0}^{[r]} \frac{(-1)^k (\lambda)_{r-k} (2t)^{r-2k}}{k!(r - 2k)!} x^r \tag{9}$$

Finally, by comparing Equation (9) with Equation (6), it can be obtained:

$$C_r^\lambda(t) = \sum_{k=0}^{[r]} \frac{(-1)^k (\lambda)_{r-k} (2t)^{r-2k}}{k!(r - 2k)!} \tag{10}$$

In the second part of proposing a suitable mathematical apparatus, the equality $\left[t_0, t_f\right] = [-1, 1]$ is specified (for the orthogonality interval), in addition to condition $\lambda > -1/2$ (as a special case of Jacobi polynomials obtained by setting $\alpha = \beta = \lambda - 1/2$). After a complex mathematical calculation from [27,28] the following series is obtained:

$$\sum_{r=0}^{\infty} C_r^\lambda(t) x^r = \sum_{r=0}^{\infty} \sum_{m=0}^{r} \frac{(2\lambda)_{r+m}}{m!(r - m)!\left(\lambda + \frac{1}{2}\right)_m} \frac{(t - 1)^m}{2^m} x^r \tag{11}$$

where symbol $(2\lambda)_{2k}$ denotes $(2\lambda)_{2k} = 2^{2k}(\lambda)_k (\lambda + 1/2)_k$, i.e., $(\lambda)_k = \lambda(\lambda + 1)(\lambda + 2) \cdots (\lambda + k - 1)$. A recurrence relation can be presented as [28]:

$$n C_n^\lambda(x) = 2(n + \lambda - 1) x C_{n-1}^\lambda(x) - (n + 2\lambda - 2) C_{n-2}^\lambda(x) \tag{12}$$

In order to obtain a basis which will represent the functions of activation mechanisms of artificial glands and activation functions of future OENN, the following function is introduced:

$$\varphi_j(x, \lambda) = \sum_{m=0}^{r} \frac{(2\lambda)_{r+m}}{m!(r-m)!\left(\lambda + \frac{1}{2}\right)_m} \frac{(x-1)^m}{2^m}, j = 0, 1, 2, \ldots \tag{13}$$

Finally, the Gegenbauer series of the polynomials applicable for implementations within the proposed intelligent structure are:

$$
\begin{aligned}
&\varphi_0(x, \lambda) = 1, \\
&\varphi_1(x, \lambda) = 2\lambda x, \\
&\varphi_2(x, \lambda) = 2\lambda(1 + \lambda)x^2 - \lambda, \\
&\varphi_3(x, \lambda) = \frac{4}{3}\lambda(1 + \lambda)(2 + \lambda)x^3 - 2\lambda(1 + \lambda)x, \\
&\vdots \\
&\varphi_{n-1}(x, \lambda) = \frac{2(n+2)}{n-1}x\varphi_{n-2}(x, \lambda) - \frac{n+5}{n-1}\varphi_{n-3}(x, \lambda)
\end{aligned}
\tag{14}
$$

For $n$ glands, the activation mechanism within each gland (1 to $n$) will be calculated as follows:

$$
\begin{aligned}
&Act_{\deg\_1}(t) = \varphi_0(CM_{out\_1}, SM_{out\_1}) \\
&Act_{\deg\_2}(t) = \varphi_1(CM_{out\_2}, SM_{out\_2}) \\
&\quad . \\
&\quad . \\
&\quad . \\
&Act_{\deg\_n}(t) = \varphi_{n-1}(CM_{out\_n-1}, SM_{out\_n-1})
\end{aligned}
\tag{15}
$$

where $CM_{out\_i}$ and $SM_{out\_i}$ are appropriate inputs to the $i$-th gland.

### 2.4. Hormonal Mechanism

The fourth and last component of novel gland architecture is hormonal mechanism, which aims to determine the hormone concentration at each time instance, $t$. The mechanism secrets hormone concertation, $C_g$, in accordance with the following equations:

$$C_g(t+1) = \beta_g C_g(t) + R_g(t+1) \tag{16}$$

$$R_g(t) = \frac{\alpha_g}{1 + C_g(t-1)} Act_{\deg}(t) \tag{17}$$

Equations (16) and (17) are introduced from [17], and showed to be an effective method for calculating hormone concentrations of artificial glands. The main difference between the two approaches is in calculating the stimulation parameter $R_g(t)$ that is based on the introduced $Act_{\deg}$ parameter as the main feature that affects the stimulation volatility. The last two unexplained parameters from Equations (16) and (17) are $\alpha_g$ and $\beta_g$ that represent the stimulation rate and the decay rate, respectively. Their values are always limited in the range [0, 1].

### 3. Implementation of AOG Network Mechanism within OENNs

#### 3.1. Basics of OENNs

The OENN network used in this research is based on the network structure proposed in [17]. The difference between the two approaches in calculating network output $y(t)$ is in the existence of hormone sensitivity parameter, $S_j$. In [17], $S_j$ is introduced to apply sensitivity features of excitatory and inhibitory glands. In this novel work, the algorithm is simplified, different gland structure (excitatory and inhibitory glands) does not exist, and only one gland unit (AOG mechanism) is introduced. This setup results in excluding

parameter $S_j$ from the mathematical apparatus, which is performed by making $S_j = 1$ for all proposed cases. That enables the network to calculate the output as follows:

$$y(t) = \sum_{i=1}^{n} \left( \omega_i(t) C_g(t) \right) \phi_i(X) + R\left(X, m, C_g(t)\right) \tag{18}$$

where $X$ is the input value, $\omega_i$ is the $i$-th network weight coefficient, $C_g$ is the concentration of hormones of a single gland, $\phi_i$ is an orthogonal function (processing element), $t$ is a time instance, and $R(X, m, C_g(t))$ represents an expansion error, which is decreasing when the number of expansion terms $m$ is increasing. The main difference between [17] and the novel network is building the orthogonal part of the structure. In [17] Chebyshev functions were used, while in this paper, previously described Gegenbauer functions will be utilized.

### 3.2. Networking of Glands

A complete structure of networked glands and integration of formed structure within the OENN system is presented in Figure 2. In the continuation of this section, how the configuration from Figure 2 is created and what components are introduced for that purpose will be explained.

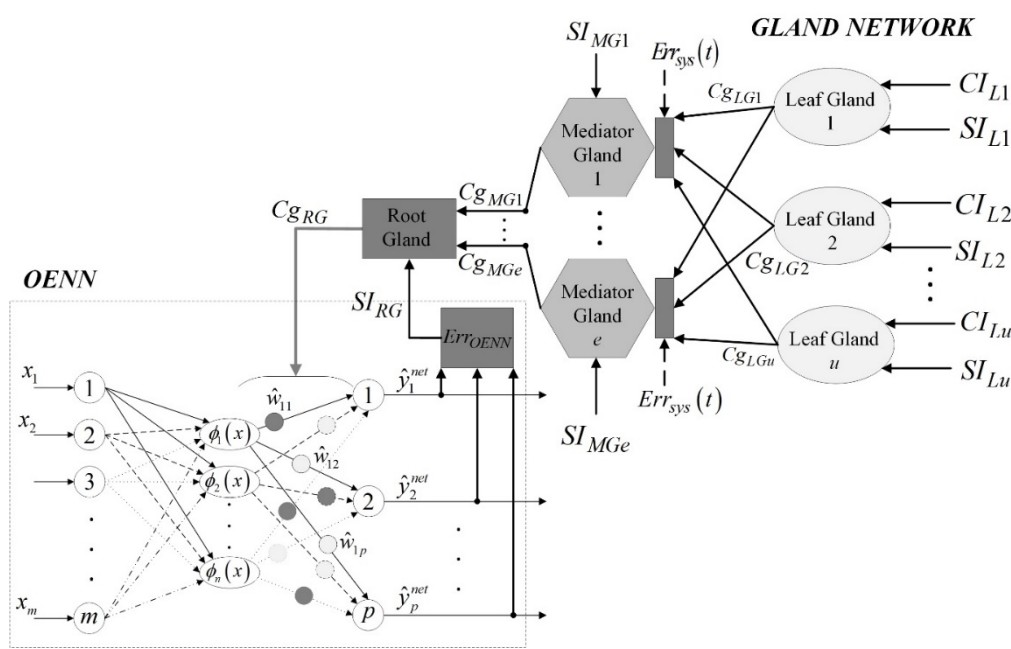

**Figure 2.** Integration of a gland network within OENN structure.

Three types of glands are proposed in this research to provide optimal responses to various environmental stimuli: a leaf gland, a mediator gland and a root gland. However, the same gland structure and operational algorithm described in the previous section are used in all three cases. The difference between glands is their position in a gland network and is determined based on the layer they individually belong to. In order to propose the gland types, first of all, the classification of all possible environmental stimuli should be made into two categories: primary and secondary environmental stimuli. The primary environmental stimuli are those that can influence the system performance significantly and that are in direct correlation with system responses. Secondary stimuli are considered those whose influence on the system cannot considerably affect the system but are still present in the system's environment.

A leaf gland directly responds to a secondary environmental stimulus (SES). Suppose an SES signal value is larger than the threshold value specified for that SES: in that case, the leaf gland is activated, the implemented gland mechanism from the previous section is

triggered, and the processing signal activities are conducted. On the other hand, if the SES signal is smaller than the threshold, the leaf gland will be in the standby regime, resulting in the $CI_{LI}$ signal equal to 0 or 1. The maximal number of leaf glands can be equal to the number of SES. Still, that number could be less than that number, considering that multiple stimuli can be introduced to a single gland cell.

A mediator gland is used to process a primary environmental stimulus (PES) brought to the gland in the form of a $SI_{MGe}$ signal. The number of mediator glands is less or equal to the number of identified PESs. Following Figure 2, additional inputs to every mediator gland are hormone concentration signals ($Cg_{Lg1}$, $Cg_{Lg2}$, ... , $Cg_{Lgu}$) introduced from the leaf glands. To accept these signals, a threshold of each mediator gland should be set, and mathematical apparatus from the previous section applied. Unlike leaf glands, mediator glands must always be in the active state. If the leaf glands' outputs are zero (if SESs do not exist or are negligible), it will not be allowed for mediator glands' *CIs* signals to be zero or insufficient to reach the gland threshold. In other words, the mediator glands must always be activated by setting them up to one of the three values: *CI* = 2, *CI* = 3 or *CI* = 4. In order to provide an optimal control input (*CI*) to a mediator gland, the Algorithms 1 is implemented.

---

**Algorithms 1** Control input selection process

---

Initial setup of all control inputs of the mediator glands is *CI* = 2
If the current system error ($Err_{sys}(t) = Y(t) - Y_{ref}(t)$) > acceptable error for $Err_{sys}(t)$
If sum ($Cg_{Lg1}$, $Cg_{Lg2}$, ... , $Cg_{Lgu}$) ≤ Mediator_Glands_Threshold
Then *CIMG* = 4 (production of hormones is intensified)
Else
CIMG = 3 (production of hormones is suppressed)
Else
*CIMG* = 2

---

Finally, all output signals of the mediator glands ($CIMG_1$, ... , $CIMGe$) are introduced to a single root gland (Figure 2). Contrary to the mediator glands, when the zero state (*CI* = 0) is not possible, that option is enabled here. If the sum of ($CIMG_1$, ... , $CIMGe$) is not enough to trigger the root gland, all PES and SES influences on a system are negligible, and there is no need for the neural network to receive any information from the AOG network. In that case, the entire AOG network will be in a standby work regime. The final input signal (*SIRG*) introduced to the root gland represents the error signal/noise signal generated during the work of a system of interest and represents the main parameter for monitoring the system's performances and accepting feedback information of the current system response. In addition to the PES signals, SIRG represents the main factor in generating appropriate hormone concentrations that will be used to influence the output layer weights of OENN.

*3.3. Training Process*

The learning algorithm of OENN is based on the algorithm proposed in [17], and implementing modified recursive least square (RLS) algorithm for updating network weights and calculating the errors:

$$e_p(t) = \sum_{i=1}^{n} \omega_i(t - \Delta t)\phi_i(t) - \hat{y}(t + n) \tag{19}$$

where $e_p(t)$ is an error calculated before the weight update process, and $n$ represents a sample rate defined by system dynamics. Further:

$$e_a(t) = \sum_{i=1}^{n} \omega_i(t)C_g(t)\phi_i(t) - \hat{y}(t + \delta) \tag{20}$$

where $e_a(t)$ is calculated error after the procedure of updating weights. In accordance with [17], the idea is that during the time, two errors converge with each other, and in the ideal case their ratio will be:

$$|e_a(t)|/|e_p(t)| \to 1 \tag{21}$$

Finally, the learning rate adaptation algorithm [17] is also exploited. Its primary purpose is to monitor the error difference sign in two consequent iterations. If the sign changes, the learning rate value decreases. On the contrary, the learning rate will increase if the sign is the same in two iterations.

## 4. Integration of OENN Structure Powered by AOG Network Mechanism within a Control Logic of a Laboratory Dynamical System

In order to exploit the proposed framework within a real-time laboratory environment, a presentation of the approach of integrating OENN with AOG Network (OENN + AOG) within an existing control logic of a laboratory dynamical system, is required. The implementation approach is presented in Figure 3.

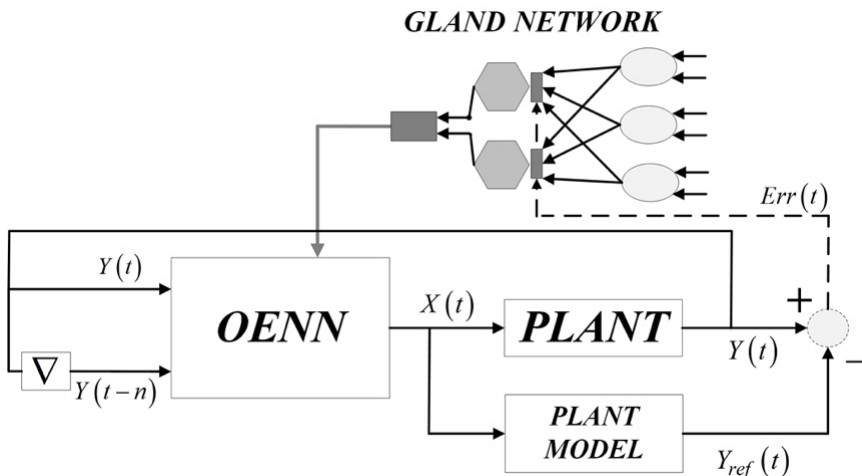

**Figure 3.** Implementation of OENN + AOG within a plant environment.

To implement the structure appropriately, all identified environmental stimuli should be categorized as SES or PES. The number of SES (l) defines the number of leaf glands, where one leaf gland is required for every unique SES. A similar procedure is required for PES, where each identified PES should be represented by one mediator gland. Another implementation option is to use one gland for multiple identified stimuli, but for accuracy and simplicity reasons, one gland per stimulus will be used for further implementation.

After identifying the number of mediator glands (*m*), it is necessary to create a fully connected network of glands with three layers and with number of nodes, l-*m*-l. One is the number of nodes in the last layer, and by default, this layer always includes just a single root gland. The output root gland signal should be introduced to the output layer weights of OENN, in accordance with Equation (18). In order to form PES and SES, the relevant sensing equipment is required to collect environmental data, process it, and introduce it to the gland network.

OENN structure should be defined following the number of input signals that should be processed and the number of output control signals that must be introduced to a plant. By default, OENN possesses two inputs introduced from the system of interest: output signal *Y(t)* and delayed signal *Y(t − n)*. If desired performances of the system are not achieved, the number of inputs can be increased, and additional delayed signals can be introduced to the network. Once the number of network inputs and outputs is specified, the number of hidden layer nodes and required setup can be established following the

procedure from Section 2.3.1 and introducing a proper number of Gegenbauer orthogonal functions.

Finally, the generated output(s) of the OENN *X(t)* is introduced to the plant input and the input of the developed plant model. The plant model should be a mathematical representation of a real-time system that will accurately describe system dynamics. The model process is in parallel signal *X(t)*, and represents simulation verification of work of a real plant. In the end, the real-time output of the plant with the model's output is compared. The error is found and introduced in front of mediator glands as a trigger component for their activation (Section 3.2).

## 5. Case Study: Magnetic Levitation System

Inteco magnetic levitation system (MLS) [29] is an open-loop unstable and nonlinear laboratory system for demonstrating phenomena of magnetic levitation (Figure 4). Levitation techniques are used in practical applications such as transportation, contactless melting, magnetic bearings, etc. One successful approach of providing optimal control of a multivariable magnetic levitation system is presented in [30]. In general, the levitation principles are based on digital and analogue solutions to maintain a steel ball in an electromagnetic field. MLS elements are the electromagnet, the hollow steel sphere, the position sensor, and the computer interface board with appropriate connecting cables. It is a single degree of freedom control system with time-varying dynamical characteristics.

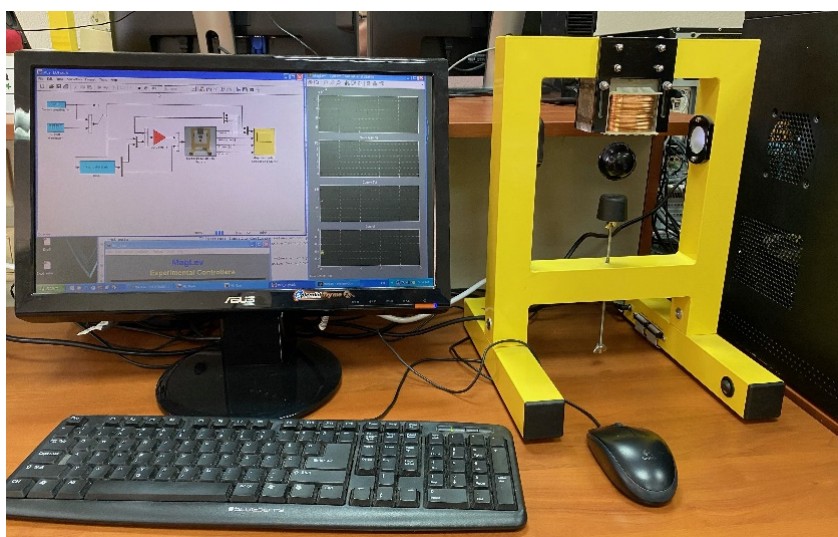

**Figure 4.** Laboratory system of MLS.

For levitation purposes of the laboratory model, a real-time controller is required. In order to keep a levitating sphere in a desirable position, the equilibrium stage of the gravitational and electromagnetic force is required. MLS operates the voltage to the electromagnet to enable the object to levitate where the levitating object's position is determined in every time step (0.01 s) through the sensor. The measured voltage corresponds to the sphere distance. Additionally, the coil current, as another measurable variable, can help explore control strategies. All control experiments of MLS can be executed in real-time in the Matlab software package, specifically Simulink Toolbox. The software control logic can be used for online identification of processes, modelling of control systems, and all design and simulation phases. Finally, the implementation of new control algorithms can also be performed in real-time.

MLS can be mathematically described in the following format [29]:

$$\dot{x}_1 = x_2$$
$$\dot{x}_2 = -\frac{F_{em}}{m} + g$$
$$\dot{x}_3 = \frac{1}{f_i(x_1)}(k_i u + c_i - x_3)$$
$$F_{em} = x_3^2 \frac{F_{emP1}}{F_{emP2}} \exp\left(-\frac{x_1}{F_{emP2}}\right)$$
$$f_i(x_1) = \frac{f_{iP1}}{f_{iP2}} \exp\left(-\frac{x_1}{f_{iP2}}\right)$$

(22)

where, $x_1 \in [0, 0.016]$, $x_2 \in \Re$, $x_3 \in [x_{3MIN}, 2.38]$, $u \in [u_{MIN}, 1]$. The electromagnetic force ($F_{em}$) depends on the distance of ball from the electromagnet and the electromagnetic coil current.

## 6. Experimental Verification

### 6.1. Control Logics Used for Experimental Comparisons

In order to verify the performance of the proposed model in a laboratory environment, four different control logics were used. The first was a default logic integrated with the MLS model, linear quadratic tracking control logic [29]. The next was based on dynamical neural networks, the approach that showed significant performance in control of nonlinear dynamical systems [30]. The third was the OEI controller, the previous proposal of the authors of this paper [17], a hybrid structure based on the combination of OENN and OEANFIS that showed a significant performance in the control of nonlinear systems weighted with notable disturbances. Finally, the last was the model proposed in this new research, OENN + AOG structure. A brief explanation of all four control logics that will be used in this section, follows.

#### 6.1.1. Linear Quadratic Tracking Control

The default tracking control of the MLS system used in Matlab is Linear Quadratic (LQ) control (Figure 5). The control and tracking policy is built for an equilibrium point. For each value of a ball position, the current values are recalculated: velocity, coil current and the control signal. All needed information about the mathematical apparatus of LQ controller can be found in [29].

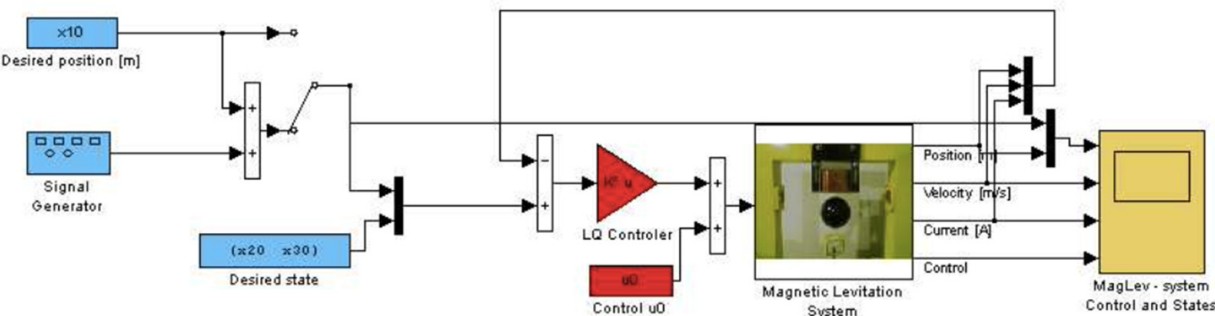

**Figure 5.** LQ tracking control diagram in Simulink environment.

#### 6.1.2. DNN

DNN structure from [31] was used for comparative purposes because of its efficient control of linear and non-linear MIMO systems. A modelled system is controlled by the proposed adaptive controller, where the algorithm uses the system outputs to tune the controller's parameters. Another reason for using DNN intelligent controller is the capability of the mechanism to cancel disturbances utilizing an adaptive filter. Dealing with disturbances was one of the main goals of this overall research paper and the OENN + OAG system, and it would be informative to see a comparison of performances achieved.



### 6.1.3. OEI Controller

The following reference model is the OENN-OEANFIS structure proposed in [17]. Instead of Gegenbauer orthogonal functions, the model from [17] is based on the utilization of Chebyshev functions. Unlike this novel research where the OENN network is used as a basis, in [17] the hybrid combination of OENN and OEANFIS was exploited. The next difference is the usage of artificial glands, where [17] used a combination of excitatory and inhibitory glands, while the AOG mechanism is exploited here. The OENN algorithms' training in both networks is similar, expanded with the learning rate adaptation algorithm. The environmental stimulus used for completing the OEI structure was pressure, measured valuable by utilizing sensing equipment.

### 6.1.4. OENN + AOG

All needed information about the intelligent control algorithm is provided in detail through Sections 2–5 of this paper.

### 6.2. Implementation of Control Logics

LQ control logic was implemented as presented in Figure 5, and no further modifications were made. All three intelligent control logics were implemented to be as similar as possible in order to be comparable and in accordance with Figure 3. The similarity was achieved by putting the same number of neurons within three neural networks and selecting the same activation functions and initial values of weights. The number of neurons within the networks was determined empirically by trying the OENN + AOG structure with hidden layer neurons from 2–200. The best performances were achieved with 40 neurons in the hidden layer. The control problem determined the number of neurons in the input layer, and it was two for the MLS problem. The number of output neurons was 1. According to the determined structure, all three intelligent control logic were based on neural networks with the structure 2-40-1.

Finally, the OENN+OEANFIS model and OENN + AOG structure required environmental stimuli for employing artificial glands. For that purpose, the following environmental stimuli were used. For OENN + OEANFIS, there was a single environmental stimulus used, a temperature. The temperature was measured by using an appropriate laboratory sensor. On the other hand, the point of OENN + AOG is to exploit numerous environmental stimuli in order to provide all relevant information to the control logic. To achieve that, the following stimuli were measured: magnetic field detection, temperature, humidity, vibration and the concentration of combustible gas in the air.

The complete process of designing the OENN + AOG structure was performed using the developed algorithm in Sections 2–4 of this paper. To integrate the AOG mechanism, it was necessary to classify all determined stimuli into two categories, which was performed in the following way: the strength of magnetic field and vibration were classified as PES (two mediator glands), while temperature, humidity and the concentration of combustible gas were classified as SES (three leaf glands). Summarized, the AOG structure was in the format (3 leaf glands–2 mediator glands–1 root gland). At that moment, the classification of all introduced environmental parameters was a subjective process and was based on human evaluation of available stimuli and determination of their importance. In future work, the authors will try to establish a universal automatic approach for classifying all relevant environmental stimuli in two categories in a reliable manner.

### 6.3. Learning Phase

The data set exploited to train the proposed structure was created by collecting real-time data from the actual work of the MLS system. The data included a reference signal that determined the desired position of the levitation object in each time step and correlated control signal that the default LQ controller generated. The sample time of exploited signals was pre-defined by the manufacturer of the system; it was 0.01 s by default, and empirically was determined that 10,000 samples per variable (100 s of measurements) would provide

optimal results. Following Figure 3, a minimal number of inputs was two, a specified input signal was $Y(t)$, and a delayed version of the specified input signal was $Y(t - n)$. Following that rule, for building the dataset for this research, $Y(t)$ represented the position signal, while for the delayed signal $Y(t - n)$ used delayed signal $Y(t)$ with a delay equal to 1, or $Y(t - 1)$ was formed. The recorded control signal of the default LQ control logic was used as the training output of the neural network. After building the set of training data for performing training processes of the OEI controller, DNN, and OENN + AOG networks, the training set was split in the following manner: 70% of the data were used for the training, 15% for validation purposes, and 15% for test cases. The test set was used to evaluate the performance of the trained model, while the validation set was used to follow the convergence of the error signal toward zero. If the validation error was equal in six consequent iterations, the network could not improve further, and the training stopped.

### 6.4. Experimental Results

Laboratory verification of the proposed structure was performed on real-time testing using the described MLS model. Tracking performances of four introduced control logics were recorded and compared simultaneously. Performances were evaluated through the position signals that individual control systems achieved and their comparisons with the reference tracking signal were assumed to be optimal. In an ideal case, a tracking position achieved due to a controller should overlap entirely with the reference signal. However, in a real-world environment, this is almost an impossible task. Figures 6 and 7 show the tracking performances of four tested control logics.

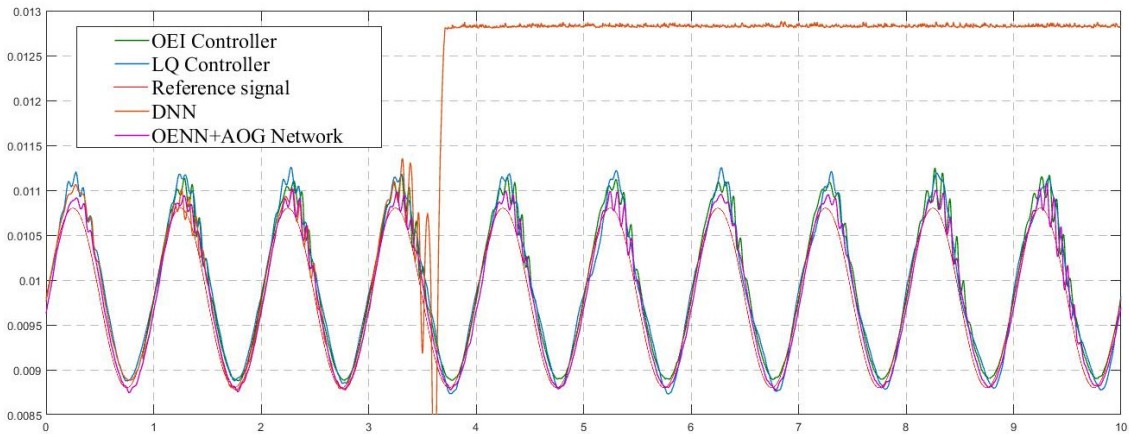

**Figure 6.** Tracking performances (position measurements).

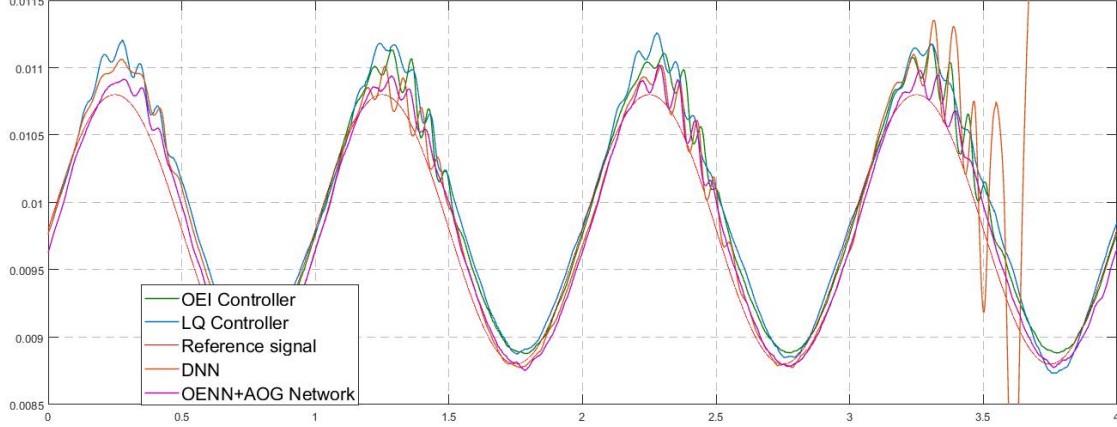

**Figure 7.** Enlarged view of part of the graph of tracking performances (position measurements).

## 7. Discussion Section

### 7.1. Achieved Performances

From Figure 6, it can be concluded that three of the four control logics successfully performed levitation of the ball for the specified trajectory. Only the DNN control algorithm could not provide tracking performances in multiple repeated experimental cases: the levitation processes stopped in every attempt, and the ball fell to the ball holder (position 0.0128 on the y axis occurred after 3.7 s). Figure 7 presents an enlarged view of part of Figure 6 and provides better insight into the difference between the other three tracking signals compared with the reference signal. Finally, the numerical presentation of achieved results from Figure 6 and summarized results are presented in Table 2. The root mean square error (RMSE) was used to calculate the overall measure of achieved performances, keeping in mind it was calculated as a difference between the measured output position MLS signal and the desired signal (the reference signal).

**Table 2.** Summarized tracking performances.

| Control Algorithm | Training Time (s) | Tracking RMSE |
|---|---|---|
| OEI Controller | 143.260 | 1.4730 |
| LQ Controller | - | 1.5949 |
| DNN Controller | 58.236 | 1.8301 |
| OENN + AOG Controller | 98.976 | 1.4761 |

As can be concluded from Table 2, OENN + AOG provided similar results to the OEI controller, with a significant reduction in computational costs expressed through the training time. In comparison to the OEI controller, OENN + AOG provided similar performances with a simpler structure and lower computational costs. Further, the OEI controller was based on the combination of online and offline learning and required special attention for setting OEANFIS. Finally, both examined structures showed smaller RMSE from the default LQ control. As final insights from conducted experiments, Figure 8 presents control signals generated from appropriate controllers. Figure 8 also presents unstable control from the DNN controller, resulting in uncontrolled oscillations and levitation process interruption. Additionally, it shows three successful control signals which enabled stable ball levitation.

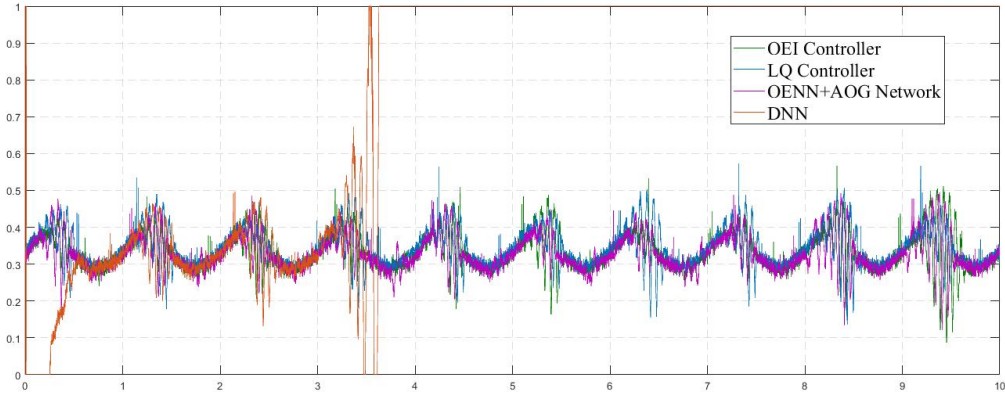

**Figure 8.** Control signals.

### 7.2. Stability Considerations

To analyze the stability of the dynamic system and proposed OENN + AOG control logic, the MATLAB LMI Control Toolbox maybe utilized [32]. LMI stability criteria are a powerful tool for matrix operations based on linear matrix inequality techniques. If there is a solution to the matrix inequalities, then the MLS system (and the corresponding intelligent control algorithm) can be considered stable [33]. However, it is challenging to

check the algebraic inequality method due to numerous adjustable network and gland parameters considering a lack of information on how to tune these parameters in every new training iteration. The exceedingly complex LMI-based stability results are valuable only for numerical purposes, and the theoretical meaning is deprived.

Further, the stability of OENN + AOG intelligent control without global feedback may be analyzed layer by layer. A single Lyapunov function can be used for each layer and generated in independent runs of the LMI solver. For example, for OENN with one hidden layer without global feedback, it should be determined if the layer is stable. This can be performed by running the LMI solver to solve LMI for this layer alone. If a solution is feasible, the stability of other hidden layers or the output layer can be analyzed. Due to the non-zero connection matrix between the layers and non-zero outputs of the first layer at its equilibrium, biases of the second layer should be adjusted accordingly before invoking the LMI solver for another layer. If a solution for the first layer is not feasible, then nothing can be concluded about the stability of the second layer either. Generalization to cases with more than two layers is straightforward. The LMI method can easily incorporate adjustable parameters into stability criteria and decrease conservativeness. In the sense of total performance evaluation of the desired results, LMI-based results are considered very effective.

Finally, the more complex the stability condition is, the less it will influence physical and theoretical meaning for analysis and will become useful only for numerical purposes. It can further lead to challenges in comparing different conditions among the LMI-based stability results. Therefore, the efficiency of exploited stability conditions can only be compared by means of a specific MLS example and not in a universally analytical way.

### 7.3. Significance of Proposed Control Logic

The contribution of the proposed algorithm can be described from two perspectives: scientific and engineering. Scientific contribution means making another step forward toward new orthogonal endocrine neural networks that have shown significant control potential in the last few years. It was shown that Gegenbauer orthogonal functions could be as equally efficient as other already utilized functions, creating a suitable environment for further considerations of different functions' applicability. Further, artificial glands were networked within a single structure (AOG structure) and tuned in a manner to treat appropriate environmental stimuli with different levels of importance for a controlled system. Finally, a new type of artificial glands was proposed, with a specially tailored mechanism within it. It was shown that such a gland could be made as an adjustable unit, where each sub-mechanism could be tailored for a specific use case. Conversely, the engineering and practical contribution of the research is in the potential to implement a similar structure for control of real-world nonlinear dynamic systems. A significant value resides in the possibility of processing an unlimited number of environmental signals, noises and disturbances with the proposed architecture. The structure is upgradable, and every new noticed environmental factor can be easily added to the existing OENN + AOG control logic. That way, the control signal in each time step is modified based on the current system's factors and environmental factors, enabling robust control and stable system work.

### 8. Conclusions

This paper presents a novel intelligent structure to control nonlinear dynamical systems. The structure is based on combining one type of orthogonal neural network (ONN) with a specially developed artificial orthogonal gland (AOG) network. The ONN used in the research is a three-layer network where the main, hidden layer is built from a series of Gegenbauer orthogonal polynomials. Each neuron has its unique polynomial. The output signals of AOG network influence output layer weights of the network. This network is a three-layer network built by combining multiple artificial glands within its structure. The gland used in this research is a new gland model, especially developed to optimize the process of collecting and processing environmental stimuli.

The AOG mechanism of a single gland includes four sub mechanisms: control, signal, activation and hormonal mechanism. This concept aims to acquire control and signal input to the gland, process it through these four sub mechanisms and generate a calculated hormone concentration. In order to enable the processing of a large number of environmental stimuli, the AOG network is made by combining single AOG mechanisms within one unit. To achieve that, the AOG network is organized into three layers: leaf layer, mediator layer and root layer. Leaf and mediator layer glands receive the input signals from the environment in the following way. First, environmental stimuli are manually divided into primary environmental stimuli (PES) and secondary environmental stimuli (SES). PES are stimuli of primary interest and importance for the network and system control. At the same time, SES includes signals that do not influence the system dynamics significantly, but still exist in the environment and could be counted as relevant.

Further, the leaf layer receives SES signals and possesses the number of leaf glands equal to the number of introduced SES stimuli. The number of mediator glands is equal to the number of PES signals. Leaf glands and mediator glands are fully connected, meaning that each leaf gland is connected to every mediator gland. Finally, all the outputs of mediator glands are introduced to the single root gland.

In order to verify the proposed intelligent structure, the laboratory model of the magnetic levitation system (MLS) was used as a complex nonlinear dynamical system. ML is characterized by significant sensitivity to disturbances and environmental conditions, and it is an excellent challenge for testing new control logics. Three additional control logics were exploited for verification purposes: default linear quadratic (LQ) tracking control logic, dynamical neural network (DNN) and the OENN + OEANFIS structure proposed in the previous research by the authors of this paper. The goal of the control systems was to track the referenced levitation signal and maintain the still ball levitation in specified positions. The OENN + AOG structure showed tracking performances similar to the more complex OENN + OEANFIS structure but with significantly less computation cost and training effort needed to achieve these results. The new structure, that relies on acquiring and processing all measurable environmental stimuli, was useful for controlling nonlinear systems and presenting the foundation for building intelligent control logic in a costless manner. Future research attempts will show its potential for the control of other linear and nonlinear systems and making an automatized approach for classifying primary and secondary environmental stimuli and selecting the proper structure of the AOG network.

**Author Contributions:** Conceptualization and methodology, M.M.; MATLAB software, M.M. and M.S.; formal analysis, S.N. and A.D.; resources, S.N. and A.O.; data curation, writing—original draft preparation, M.M. and A.O.; writing, A.D. and M.M.; visualization, A.D. and M.S.; supervision, S.N. and M.M.; project administration and funding acquisition, A.O. All authors have read and agreed to the published version of the manuscript.

**Funding:** The authors disclosed the receipt of the following financial support for the research, authorship, and/or publication of this article: This research received funding from the Shift2Rail Joint Undertaking under the European Union's Horizon 2020 research and innovation program under Grant Agreement No. 881784.

**Institutional Review Board Statement:** Not applicable.

**Informed Consent Statement:** Not applicable.

**Data Availability Statement:** Not applicable.

**Acknowledgments:** This work has been supported by the Ministry of Education, Science and Technological Development of the Republic of Serbia. Special thanks to Serbian Railways Infrastructure, and Serbia Cargo for their support in conducting the SMART and SMART2 OD field tests.

**Conflicts of Interest:** The authors declare no conflict of interest.

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
