# Peer review of "An Approach to Networking a New Type of Artificial Orthogonal Glands within Orthogonal Endocrine Neural Networks"

_applsci, doi:10.3390/app12115372_

Round 1

Reviewer 1 Report

In general, the paper is well written and organized. It concerns the practical issues covering the broadly understood Artificial Intelligence paradigm. A set of innovative and comparative results derived from the laboratory magnetic levitation system seems to be accurate and worth presenting to a scientific community. However, in order to slightly improve a correctness of the paper,  some small drawbacks should immediately be brushed as follows:

  1. In a number of places, the Authors have written about the time-varying dynamical properties as well as the adaptive mechanism.  I do not suppose that the examined real-life model of MLS is time-variant. The discussed issue should be explained more clearly.
  2. The convergence and stability behaviors of the proposed numerical approach should be explored deeply as well.
  3. Since the Authors have cited various references, it would be appreciated to tabulate their own original achievements.
  4. The position https://www.mdpi.com/2076-3417/12/2/674 should  increase the readability of the submitted paper.

Author Response

Dear reviewers, we would like to thank you for your valuable comments and meaningful observations. After revising the paper according to your requirements, we are sure you helped in improving our paper significantly. Answers to your questions are provided below and related modifications within the paper are introduced appropriately.

Reviewer 1

  1. In a number of places, the Authors have written about the time-varying dynamical properties as well as the adaptive mechanism. I do not suppose that the examined real-life model of MLS is time-variant. The discussed issue should be explained more clearly.

Answer: The time-varying specifics are now explained more clearly in Page 4.

  1. The convergence and stability behaviors of the proposed numerical approach should be explored deeply as well.

Answer: Stability considerations are now included within the Discussion section, subsection 7.2, page 19.

  1. Since the Authors have cited various references, it would be appreciated to tabulate their own original achievements.

Answer: The table with systematized previous authors' results in the field of OENNs and artificial endocrine influences is now given within a new subsection 1.2, Page 2.

  1. The position https://www.mdpi.com/2076-3417/12/2/674 should increase the readability of the submitted paper.

Answer: The position is added to the paper (Page 14).

Reviewer 2 Report

The article is devoted to developing an intelligent control structure based on an orthogonal endocrine neural network and a network of artificial orthogonal glands. The developed orthogonal endocrine neural network uses orthogonal Gegenbauer functions as the main elements of hidden layer neurons. The orthogonal endocrine neural network is connected to the artificial orthogonal gland network through the output layer weights of the orthogonal endocrine neural network. The network of artificial orthogonal glands is built from three levels: the end level, the mediator level, the root level, and the unified type processing unit. The working mechanism of each artificial orthogonal gland includes four sub-mechanisms, the purpose of which is to process the input data and create the proper output concentration of artificial hormones. All environmental incentives are researched and classified into primary and secondary environmental incentives. Primary stimuli are introduced into the mediator glands, and secondary stimuli are introduced into the leaf glands. The leaf and mediator layers are fully connected. As a result, the outlets of the mediator layer are introduced into the root gland. The proposed design has been tested in laboratory conditions on a laboratory magnetic levitation system. The authors compared the obtained results with two additional intelligent algorithms and standard linear-quadratic control logic. The advantages of the orthogonal endocrine neural network + artificial orthogonal gland network structure are improved tracking performance compared to traditional linear-quadratic control, better adaptability to environmental conditions and the potential to obtain optimal results with a more straightforward structure with lower computational cost compared to other intelligent structures.

Despite the satisfactory quality of the article, some shortcomings need to be corrected.

  1. The abstract should be rewritten, including a brief description of the research actuality and results obtained by the authors.
  2. The aim of the paper should be defined.
  3. The methods proposed by the authors should be separated from well-known ones.
  4. The argumentation of the neural network architecture is needed.
  5. The dataset used in the experimental study should be described in more detail.
  6. The discussion section should be included with a comparative analysis of obtained results from other research.
  7. The scientific and practical novelty of the research should be highlighted.

In summarizing my comments, I recommend that the manuscript is accepted after minor revision. 

Author Response

Dear reviewers, we would like to thank you for your valuable comments and meaningful observations. After revising the paper according to your requirements, we are sure you helped in improving our paper significantly. Answers to your questions are provided below and related modifications within the paper are introduced appropriately.

Reviewer 2

  1. The abstract should be rewritten, including a brief description of the research actuality and results obtained by the authors.

Answer: The abstract is rewritten in accordance with the recommendations.

  1. The aim of the paper should be defined.

Answer: The main aims of the paper are now more clearly defined and provided on Page 3.

  1. The methods proposed by the authors should be separated from well-known ones.

Answer:  Methods and algorithms that the authors of the paper previously proposed are now separated from other references and summarized in subsection 1.2 and tabulated in Table 1 (Page 2).

  1. The argumentation of the neural network architecture is needed.

Answer: The argumentation for using the proposed architecture is now given on Page 4.

  1. The dataset used in the experimental study should be described in more detail.

Answer: A more extensive description is now added to page 17.

  1. The discussion section should be included with a comparative analysis of obtained results from other research.

Answer: The discussion section is now created within the paper (Page 18), including three subsections that cover an explanation of achieved results, stability considerations, and the significance of the proposed control logic.

  1. The scientific and practical novelty of the research should be highlighted.

Answer: Novelties of the paper are clearly added to the discussion section, subsection 7.3 (page 20)